# Collective magnetism in an artificial 2D XY spin system

Naëmi Leo [1,2], Stefan Holenstein[3,4], Dominik Schildknecht[1,2,5], Oles Sendetskyi [1,2], Hubertus Luetkens[3], Peter M. Derlet[5], Valerio Scagnoli[1,2], Diane Lançon[1,2,6], José R.L. Mardegan[7], Thomas Prokscha [3], Andreas Suter[3], Zaher Salman [3], Stephen Lee[8] & Laura J. Heyderman[1,2]

Two-dimensional magnetic systems with continuous spin degrees of freedom exhibit a rich spectrum of thermal behaviour due to the strong competition between fluctuations and correlations. When such systems incorporate coupling via the anisotropic dipolar interaction, a discrete symmetry emerges, which can be spontaneously broken leading to a low-temperature ordered phase. However, the experimental realisation of such two-dimensional spin systems in crystalline materials is difficult since the dipolar coupling is usually much weaker than the exchange interaction. Here we realise two-dimensional magnetostatically coupled XY spin systems with nanoscale thermally active magnetic discs placed on square lattices. Using low-energy muon-spin relaxation and soft X-ray scattering, we observe correlated dynamics at the critical temperature and the emergence of static long-range order at low temperatures, which is compatible with theoretical predictions for dipolar-coupled XY spin systems. Furthermore, by modifying the sample design, we demonstrate the possibility to tune the collective magnetic behaviour in thermally active artificial spin systems with continuous degrees of freedom.

[1] Laboratory for Mesoscopic Systems, Department of Materials, ETH Zurich, 8093 Zurich, Switzerland. [2] Laboratory for Multiscale Materials Experiments, Paul Scherrer Institut, 5232 Villigen PSI, Switzerland. [3] Laboratory for Muon Spin Spectroscopy, Paul Scherrer Institut, 5232 Villigen PSI, Switzerland. [4] Physik-Institut der Universität Zürich, Winterthurerstrasse 190, 8057 Zurich, Switzerland. [5] Condensed Matter Theory Group, Paul Scherrer Institut, 5232 Villigen PSI, Switzerland. [6] Laboratory for Neutron Scattering and Imaging, Paul Scherrer Institut, 5232 Villigen PSI, Switzerland. [7] Swiss Light Source, Paul Scherrer Institut, 5232 Villigen, Switzerland. [8] School of Physics and Astronomy, SUPA, University of St. Andrews, St Andrews KY16 9SS, UK. Correspondence and requests for materials should be addressed to N.L. (email: naemi.leo@psi.ch)

According to the Mermin–Wagner theorem, two-dimensional magnetic systems with continuous global symmetry are not expected to exhibit long-range order at finite temperatures[1]. A well-known example is the XY-rotor model, where magnetic moments interact via an isotropic Heisenberg interaction, and it is only quasi-long-range order that emerges at a Berenzinskii–Kosterlitz–Thouless transition[2,3]. Introduction of anisotropic dipolar interactions, however, breaks the continuous symmetry of the system and the Mermin–Wagner theorem no longer applies. However, regardless of the discrete symmetry of the Hamiltonian, two-dimensional lattices of dipolar-coupled XY moments (referred to as dXY systems) still feature a continuous ground-state degeneracy, which reflects the frustration due to competing ferromagnetic and antiferromagnetic contributions of the dipolar interaction[4,5]. Here, it is the thermal fluctuations that select a limited number of spin states from the continuous manifold and lead to a long-range ordered phase at finite temperatures[5–7]. Such an order-by-disorder transition[8,9] is a rare example in which the magnetic phases in purely classical spin systems are selected by entropic rather than energetic contributions. For interacting dXY moments on a square lattice, a continuous transition to antiferromagnetic stripe order is predicted[5–7,10,11]. The universality class of this transition has not yet been fully resolved, although both Ising[5,12] and XYh4[13,14] universality are under discussion. Therefore, magnetostatically coupled spin systems (including dipolar and weaker higher-multipole interactions) with continuous degrees of freedom are predicted to feature intriguing physics.

Experimentally, two-dimensional dXY systems have remained largely unexplored since, in microscopic systems, dipolar coupling is usually significantly weaker than short-range exchange interactions. The dipolar interaction naturally dominates, however, in arrays of mesoscopic magnets created with lithographic methods, where the macrospin anisotropy and lattices can be tailored. Such artificial spin systems display a wide variety of static moment configurations, show intriguing field-driven and thermal magnetic properties, and have been used to explore emergent spin correlations in two dimensions, with a major focus on analogues of the pyrochlore spin ices[15–19]. Thermally fluctuating systems can be created by incorporating superparamagnetic nanomagnets[20–22] and the equilibrium behaviour in the thermodynamic limit can be measured[18,19].

In this work, we experimentally realise square-lattice arrays of interacting nanodiscs, acting as good approximations of dXY moments with continuous spin degrees of freedom, and measure their thermal behaviour with low-energy muon-spin relaxation and soft X-ray scattering. Careful comparison of strongly and non-interacting samples provides a means to distinguish the contributions from collective and single-particle behaviour to the muon-spin relaxation signal. For the strongly interacting samples, we observe a phase transition and the development of static magnetism at a temperature where the moments in isolated nanomagnets are still fluctuating. We further demonstrate the emergence of spatial correlations, which is compatible with the long-range order theoretically predicted for dXY spin systems, with the observation of magnetic superstructure peaks by resonant elastic X-ray scattering. This realisation of strongly coupled thermally active systems with continuous spin degrees of freedom opens the way for future investigations of dXY spin systems with a variety of arrangements that are predicted to have rich phase diagrams.

## Results

**Artificial dipolar-coupled XY spin systems**. Most previously considered artificial spin systems consist of arrays of elongated

magnetic islands with Ising-like degrees of freedom. Here we fabricate arrays of thin permalloy ($Ni_{80}Fe_{20}$) discs with electron-beam lithography (see Methods) with the aim to mimic the continuous spin degrees of freedom in magnetostatically coupled XY spin systems. Owing to their magnetic shape anisotropy with a large diameter-to-thickness ratio, the magnetisation in the thin permalloy discs will be confined within the plane. If the disc diameter is below ≈150 nm, the disc falls into a single-domain state[23–25] and, to first order, the magnetic far field is that of a point-dipole moment[26]. Corrections to this approximation are caused, e.g., by quadrupolar or higher-order moments originating from non-uniform magnetisation within the discs[27]. The effect of these higher-order magnetostatic multipole contributions can be approximated by a modified interaction energy[26].

Using these XY macrospins as basic building blocks, one can explore the influence of magnetostatic interactions on the field-driven behaviour[28–30], superparamagnetic fluctuations[22], and, as shown in this work, on the ordering at thermal equilibrium associated with continuous spin degrees of freedom.

Assuming no other anisotropies are present, the moment is free to rotate within the plane of the thin-film discs[22,23,31,32]. In this case, one would expect angular spin fluctuations down to low temperatures. However, in thin-film nanomagnets, disorder is present due to variations in magnetocrystalline anisotropy and shape, and the in-plane moment fluctuations become increasingly constrained as the temperature reduces. This results in a temperature below which the fluctuations of the individual macrospins are slower than the timescale of the experiment, which is referred to as the single-particle blocking temperature $T_B$[33–35]. Nevertheless, for small imperfections and temperatures larger than $T_B$, the magnetic moments can still explore the full in-plane angular range, so that the ensemble is a close approximation of the equilibrium state of a dXY system.

For strong magnetostatic coupling, the mutual interaction between the nanomagnets can lead to the spontaneous transition to long-range ordered equilibrium configurations, which are expected to emerge below a critical temperature $T_C$. To first approximation, this critical temperature is proportional to the dipolar coupling energy determined by the magnetic moment $m$ of the single nanomagnet as well as the lattice parameter $a$ as follows:

$$T_C \propto \frac{m^2}{a^3}. \qquad (1)$$

For the square-lattice dXY model, a second-order phase transition from a superparamagnetic to a collinear stripe-ordered state with antiferromagnetically coupled ferromagnetic spin chains along the $x$ or $y$ lattice directions is predicted[5–7,10,29]. Higher-order multipole moments, leading to additional terms for the interparticle magnetostatic coupling, neither affect the symmetry-breaking mechanism nor the expected long-range order at zero field[11,29,30], but can lead to a quantitative change of the transition temperature.

Both temperature scales, $T_B$ and $T_C$, will influence the magnetic fluctuations in extended lattices of dXY moments. To separate the contributions of collective and single-particle behaviour to the temperature-dependent magnetic behaviour, three sample sets are used for comparative measurements (Table 1). Each set consists of a strongly and a non-interacting sample. For both samples in a given set, the nanomagnets have the same disc diameter $d$ and thickness $h$, and therefore the single-particle blocking temperatures $T_B$ are the same, leaving the lattice periodicity $a$ as the only difference. The nanomagnets in the strongly interacting samples have edge-to-edge distances of 20–30 nm and the large magnetostatic interactions will result in a high value of the critical

**Table 1 Comparison of sample sets**

|  | Set 1 | Set 2 | Set 3 |
|---|---|---|---|
| **Properties of the magnetic nanodiscs** | | | |
| $d$ | 70 nm | 40 nm | 35 nm |
| $h$ | 2.7 nm | 3.5 nm | 6.0 nm |
| **Properties of the strongly interacting samples** | | | |
| $a$ | 100 nm | 70 nm | 55 nm |
| $T_C$ | 97.1 ± 1.0 K | 106.0 ± 1.2 K | 143.1 ± 4.0 K |
| $g_{fast}$ | 3.94% | 6.85% | 10.77% |
| **Properties of the non-interacting samples** | | | |
| $a$ | 200 nm | 140 nm | 110 nm |
| $T_B$ | 39.6 ± 6.3 K | 49.1 ± 8.2 K | 67.6 ± 12.5 K |

Given are nanodisc diameter and thickness, $d$ and $h$, and the lattice periodicity $a$. Values for the single-particle blocking temperature $T_B$, the critical temperature $T_C$, and the magnitude of the fast fraction $g_{fast}$ are obtained from zero-field $\mu$SR measurements

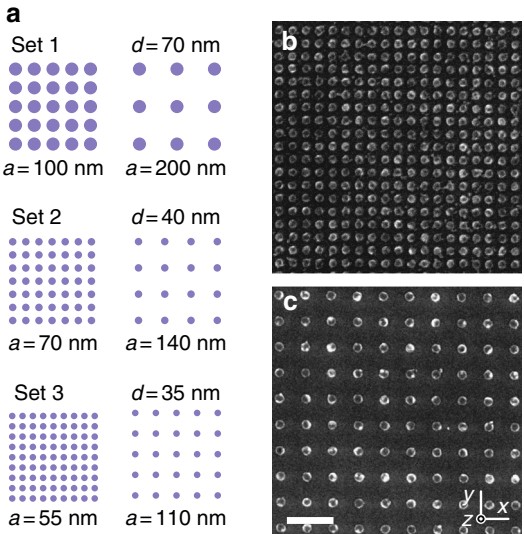

**Fig. 1** Artificial dXY spin systems. **a** Three sample sets differing in disc diameter $d$, disc thickness $h$, and lattice periodicity $a$ were fabricated with each set consisting of a strongly interacting sample (left) and a non-interacting sample (right). The schematics are depicted to scale. **b**, **c** Scanning electron microscope images of magnetic permalloy nanodiscs on a silicon substrate of samples in Set 2, consisting of **b** the strongly interacting sample and **c** the non-interacting sample. The scale bar is 250 nm. The patterned area of the samples is between 2.3 and 3 cm$^2$

temperature $T_C$. In contrast, the non-interacting samples have twice the lattice periodicity of the strongly interacting samples, effectively reducing the dipolar coupling strength in Eq. (1) by a factor of eight so that $T_B > T_C$. Therefore, the non-interacting samples are expected to only give insight into the single-particle blocking dynamics.

Scanning electron microscopy images of artificial dXY spin systems fabricated by electron-beam lithography (see Methods) are shown in Fig. 1. The samples contain $(7\text{–}90) \times 10^9$ nanomagnets, and thus can be assumed to be in the thermodynamic limit. The single-particle blocking temperatures determined by $\mu$SR experiments lie in the range from 50 to 70 K, below which superparamagnetic fluctuations are slower than the measurement timescale.

**Low-energy $\mu$SR experiments**. In this work, we measure signatures of emergent long-range order in the artificial dXY spin systems using low-energy muon-spin relaxation (LE-$\mu$SR). $\mu$SR is ideally suited to measure the temperature dependence of magnetic fluctuations in artificial spin systems due to the high sensitivity to the small magnetic fields emanating from the sample as well as to moment fluctuations with frequencies in the range from kHz to GHz, and the possibility to measure in zero magnetic field[18]. Here, low-energy spin-polarised muons are implanted close to the permalloy discs in a gold stopping layer and sample the stray fields generated by the nanomagnets. Inhomogeneities in the field distribution and magnetic fluctuations lead to a loss of the polarisation $P(t)$ of the muon ensemble[36], which gives information on the phase transitions and magnetic order present in the sample.

**Temperature-dependent muon-spin depolarisation**. To determine the equilibrium properties of artificial dXY systems, we measured temperature-dependent zero-field $\mu$SR relaxation spectra with the initial muon spin parallel to the in-plane $x$ direction of the square lattice (see Methods). The measured muon-spin polarisation function $P(t)$ is a smoothly decaying function with time and can be best described by a sum of a constant and two exponentially damped signals,

$$P(t) = g_0 + g_{slow}e^{-\lambda_{slow}t} + g_{fast}e^{-\lambda_{fast}t}, \quad (2)$$

with relative fractions $g_i$ (where $i = 0$, slow, or fast) and the corresponding depolarisation rates $\lambda_{slow} \leq \lambda_{fast}$. Details of the fitting procedure are described in the Methods section.

To distinguish the influence of the collective magnetic behaviour of the nanomagnet ensemble and single-particle fluctuations on the muon polarisation $P(t)$, comparative

measurements were performed for samples with strong and negligible coupling, respectively. Representative $\mu$SR spectra $P(t, T)$ for the strongly interacting sample with $d = 40$ nm and $a = 70$ , nm and for the non-interacting sample with $d = 40$ nm and $a$ , $= 140$ nm (Sample Set 2 in Table 1) are shown in Fig. 2.

For the non-interacting sample, there is almost no damping of the muon-spin depolarisation function at temperatures above 60 K and only moderate damping at lower temperatures (Fig. 2a). Here, the polarisation $P(t)$ can be described by a single-exponential decay, i.e. considering only two fractions in Eq. (2) (and thus $g_{fast} \equiv 0$). The extracted values for the temperature-dependent depolarisation rate $\lambda_{slow}^{n.i.}(T)$, shown in Fig. 3h, exhibit a step-like increase on reducing the temperature, which is associated with the single-particle freezing of the individual nanomagnets. A sigmoid fit to $\lambda_{slow}^{n.i.}(T)$ gives a centre of 49.1 K ± 8.2 K corresponding to the mean value of the blocking temperature $T_B$ of the magnetic ensemble measured with muons[35].

The measurements of $P(t)$ for the strongly interacting sample (Fig. 2b) show sizeable damping emerging at temperatures well above the single-particle blocking temperature $T_B$. Furthermore, around 100 K, a rapid contribution to the muon-spin depolarisation is observed for times below 0.5 μs, leading to a sizeable signal loss. This contribution is described by the term $g_{fast}e^{-\lambda_{fast}t}$ in $P(t)$ [Eq. (2)]. Interestingly, the depolarisation rate $\lambda_{fast}(T)$ shown in Fig. 3c (red diamonds) has a sharp peak around 100 K, defining a critical temperature $T_C$. The value of $T_C$ is determined by fitting a piece-wise linear function $\lambda_{fast}(T)^{-1} \propto (T - T_C)$ to the diverging signal of $\lambda_{fast}$ (red solid lines in Fig. 3c). In contrast, $\lambda_{slow}(T)$ shown in Fig. 3d (blue dots) is low at high temperatures and, coming from high temperatures, gradually increases below 100 K and saturates below 50 K.

The strong decay of the muon-spin polarisation observed in the strongly coupled dXY system, which is not seen in the non-interacting sample, implies that mutual interactions can lead to emergent correlated behaviour at temperatures, where the isolated macrospins would be still superparamagnetic.

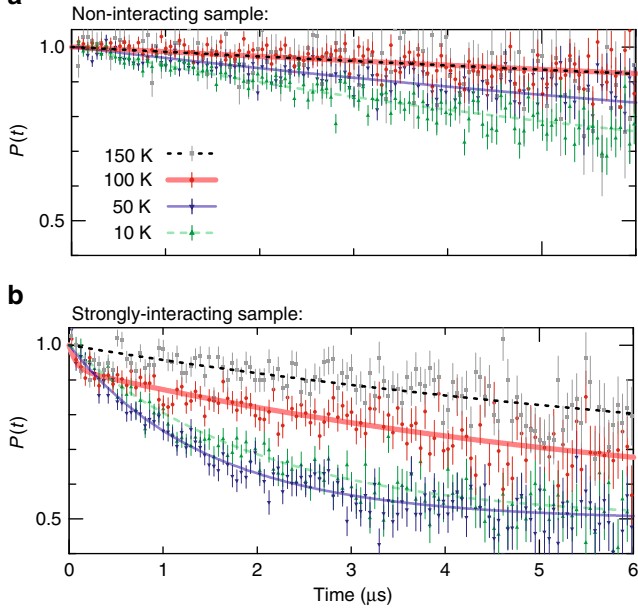

**Fig. 2** Temperature-dependent zero-field $\mu$SR depolarisation spectra for Sample Set 2. **a** Time-dependent muon-spin depolarisation function $P(t)$ (symbols) for the non-interacting sample ($a = 140$ nm), with fits according to the Eq. (2) (lines). The muon polarisation gradually decreases at lower temperatures due to the slowing down of the single-particle magnetic fluctuations. **b** $P(t)$ for the strongly interacting sample ($a = 70$ nm). In comparison to the non-interacting sample, the curves show already significant damping at higher temperatures. In addition, at $T_C \approx 100$ K a fast relaxation contribution is observed at early times (thick red curve). The error bars indicate the standard deviation, with a total number of events per spectrum of $5 \times 10^6$ (**a**) and $10^7$ (**b**)

To gain further insight into the effect of magnetic fluctuations on the muon-spin depolarisation, we performed $\mu$SR experiments on the strongly interacting sample using a longitudinal measurement geometry (see Methods). Here we apply a magnetic field that suppresses the influence of the internal static fields on $P(t)$, leaving fluctuations as the only source of muon-spin depolarisation[36]. Comparative measurements for three different temperatures are shown in Fig. 4a–c. The difference between $P(t, \mu_0 H = 0$ mT) and $P(t, \mu_0 H = 6$ mT) is associated with the contribution of the static magnetic moment configurations to the muon-spin depolarisation. We calculated the phenomenological parameter $g_{static}(T)$ from the integrated area between the curves (i.e. the shaded region in Fig. 4a–c) and, from the temperature dependence of $g_{static}(T)$ (Fig. 4d), it can be seen that static magnetism develops below 100 K. Furthermore, the fluctuation rates $f_{fluct}(T)$ (Fig. 4e) and the width of the internal magnetic field distribution $\delta(T)$ (Fig. 4f) can be determined by simultaneous fitting of the data obtained with and without applied magnetic fields with the so-called dynamic-exponential Kubo–Toyabe function[37,38]. The additional measurements shown in Fig. 4e and f, compared to those shown in Fig. 4d, are supplementary measurements obtained at 6 mT. Here, the $P(t, T, 6$ mT) curves were fitted with the same Kubo–Toyabe function but, because there are no zero-field measurements at the same temperature, the fit function is less constrained, and the errors of the fitting parameters are larger. In summary, at high temperatures we observe high fluctuation rates $f_{fluct}$ (Fig. 4e), that rapidly decrease below 100 K, and vanish below 40 K.

**Tailoring of magnetic properties by sample design.** To elucidate the role of magnetostatic coupling on the collective magnetic behaviour, we performed equivalent zero-field muon-spin depolarisation measurements for the three different sets of strongly and non-interacting samples listed in Table 1. These measurements, along with the fitted depolarisation rates are summarised

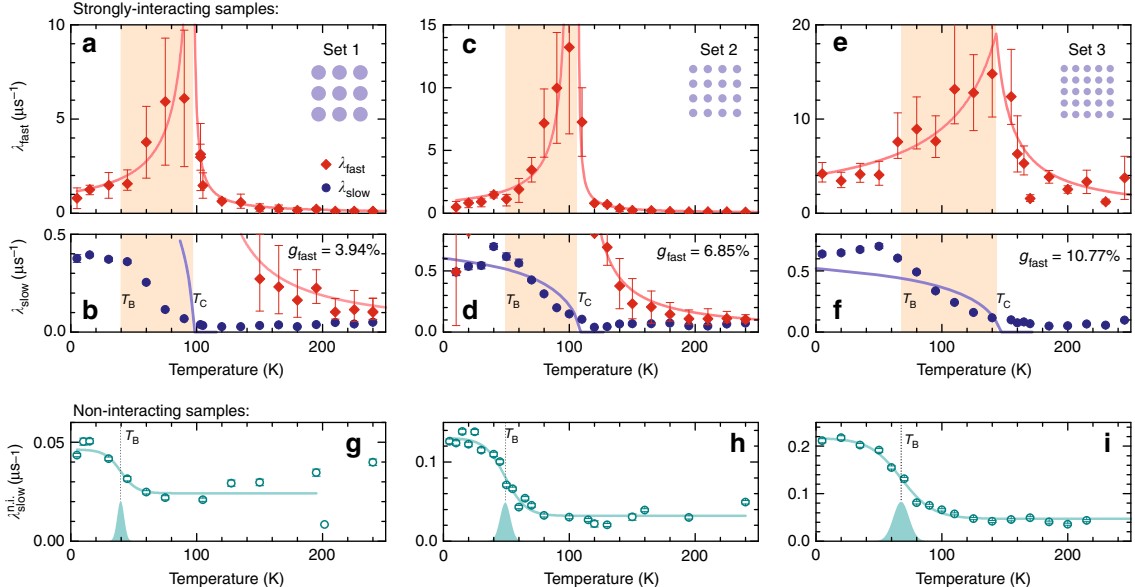

**Fig. 3** Comparison of depolarisation rates for zero-field measurements of different sample sets. **a–f** Temperature dependence of the two depolarisation rates observed in the strongly interacting dXY systems: $\lambda_{fast}$ peaks around the critical temperature $T_C$ (red diamonds), and $\lambda_{slow}$ increases below $T_C$ (blue circles). The shaded area denotes the temperature window between $T_B$ and $T_C$ where collective magnetic fluctuations can be observed. The blue solid lines in **b, d, f** denote the $\mu$SR depolarisation rate calculated from a theoretical mean-field treatment of dipolar XY spin systems (see Supplementary Note 1). The red lines in **a–e** are a fit according to $\lambda_{fast}^{-1} \propto (T - T_C)$. **g–i** Temperature dependence of the single-exponential depolarisation rate $\lambda_{slow}^{n.i.}$ for the non-interacting samples. The solid cyan curve denotes a sigmoidal function with a Gaussian distribution centred around the single-particle blocking temperature $T_B$ (shaded curve)

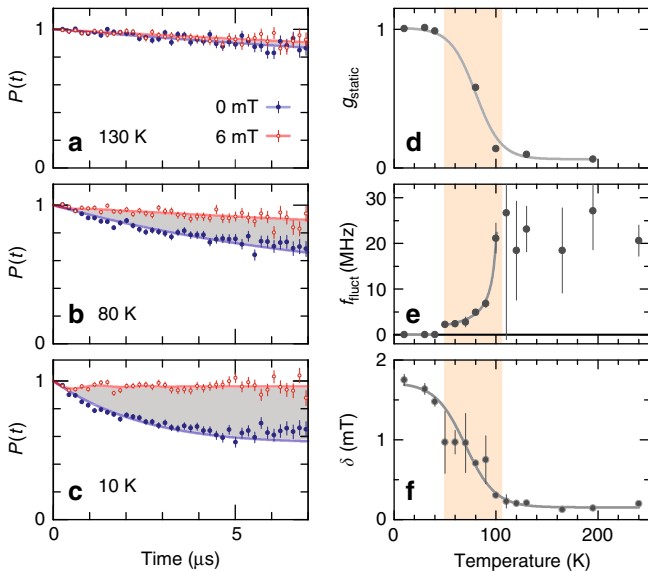

**Fig. 4** Longitudinal-field μSR measurements for the strongly interacting sample of Set 2. **a–c** $P(t)$ without field (blue dots) and with an applied field of $\mu_0 H_z = 6$ mT (red open circles) at different temperatures. The error bars indicate the standard deviation, with $10^7$ events per spectrum. **d** The parameter $g_{static}(T)$ is obtained from the integrated difference (i.e. grey area) between the spectra in **a–c**, and indicates the presence of quasi-static magnetism at low temperatures. The values of $g_{static}$ are normalised to the integrated area at lowest temperatures, i.e. $g_{static}(10 \text{ K})$. **e** Fluctuation rate $f_{fluct}(T)$ obtained from dynamic Kubo–Toyabe fits (solid lines in **a–c**). Above $T_C$ the fluctuation rate appears constant (fast-fluctuating limit) and slows down considerably below $T_C$ until the magnetic fluctuations freeze out below 40 K. **f** Width of the internal field distribution $\delta(T)$. Lines in **d–f** are guides to the eye

in Fig. 3. For the strongly interacting samples, there is an apparent divergence of $\lambda_{fast}(T)$ (red diamonds in Fig. 3a, c, e) at a critical temperature $T_C$, below which the value of $\lambda_{slow}(T)$ (blue dots in Fig. 3b, d, f) gradually increases. For the non-interacting samples, there is a step-like increase in the depolarisation rate $\lambda_{slow}^{n.i.}(T)$ (cyan circles in Fig. 3g–i), corresponding to the blocking temperature $T_B$. Below $T_B$, the values of $\lambda_{slow}^{n.i.}(T)$, $\lambda_{slow}(T)$ and $\lambda_{fast}(T)$ level off, indicating that magnetic fluctuations vanish as the moments freeze out at lower temperatures. In all strongly interacting samples, there is a temperature window between $T_B$ and $T_C$ (shaded area in Fig. 3) where collective magnetic order emerges.

We observe qualitative trends for decreasing nanomagnet diameter $d$ and lattice periodicity $a$ (i.e. going from left to right in Fig. 3): For the non-interacting samples, there is an increase in the single-particle blocking temperature $T_B$ with decreasing dot diameter $d$ (Fig. 3g–i). For the strongly interacting samples, the values for $T_C$, $g_{fast}$ and the experimental maximum values of $\lambda_{slow}$ and $\lambda_{fast}$ all increase across the series. Also, going across the series as the lattice periodicity $a$ decreases, we observe an increase in the critical temperature $T_C$, which is in line with Eq. (1).

We can conclude that there are three (two) temperature regimes for the magnetic behaviour of strongly interacting (non-interacting) dXY systems: At high temperatures the magnetic response is governed by fast superparamagnetic fluctuations, whereas at low temperatures we observe the freezing of magnetic configurations due to single-particle blocking. Between these limiting regimes, mutual interactions can modify magnetic fluctuations and spin-equilibrium configurations, leading to correlations and long-range ordered states in the considered artificial dXY spin systems. In the following we will discuss the

μSR signatures of each regime, and how they relate to the magnetic behaviour of the sample determined by the energy scales $k_B T_C$ and $k_B T_B$.

**High-temperature superparamagnetic regime**. At high temperatures, i.e. $T > T_C$ for strongly interacting, and $T > T_B$ for non-interacting samples, the superparamagnetic fluctuation rates of the single nanomagnets are much higher than the muon pre-cession frequency, i.e. the Larmor frequency. Consequently, the time-averaged magnetic field at the muon site is small (so-called motional narrowing in the fast-fluctuating regime[36,38]) leading to the small, almost-constant muon-spin depolarisation rates $\lambda_{slow}(T > T_C)$ and $\lambda_{slow}^{n.i.}(T > T_B)$.

**Intermediate correlated regime**. The temperature window $T_C \geq T \geq T_B$ (shaded areas in Figs. 3, 4), which we can define for the strongly interacting samples only, is characterised by the emergence and growth of static magnetic correlations. Experimental signatures related to this regime are the gradual increase of the muon-spin depolarisation rate $\lambda_{slow}(T_C \geq T \geq T_B)$ (Fig. 3b, d, f), the emergence of a large additional contribution to the muon-spin depolarisation $\lambda_{fast}(T)$ that peaks at $T_C$ (Fig. 3a, c, e), the rapid reduction of the fluctuation rate $f_{fluct}(T)$ (Fig. 4e, i.e. the system is slowing down), and the increase of the parameter $g_{static}(T)$ (Fig. 4d). As the fast muon-spin depolarisation described by $\lambda_{fast}(T)$ is not observed for the non-interacting samples, it cannot be caused by the presence of local disorder or single-particle behaviour. Therefore, our μSR results demonstrate that mutual interactions in strongly interacting artificial dXY spin systems lead to quasi-static correlations at temperatures where the isolated nanomagnets would still be in the superparamagnetic fluctuating regime.

We now focus on the temperature dependence of the muon-spin depolarisation rates $\lambda_{slow}(T)$ and $\lambda_{fast}(T)$ in more detail and relate this to the expected behaviour of a second-order phase transition from the paramagnetic into a stripe-ordered phase predicted for the square-lattice dXY spin system[5,30]. As the muon-spin depolarisation is both influenced by the width of the field distribution experienced by the muon ensemble as well as magnetic fluctuations within the sample, both (quasi-) static and dynamic properties of the emergent correlations can contribute to the observed μSR spectra.

To relate the static magnetic order to the measured μSR spectra, we calculated the time-dependent muon-spin depolarisation for emerging magnetic order using a mean-field description of the dXY order parameter[6,12] and randomised muon-spin stopping positions in accordance with the implantation profile in the 80 nm-thick gold layer (see Methods and Supplementary Note 1). The depolarisation function $P(t)$ obtained from these mean-field simulations yields two depolarisation rates that vanish above $T_C$ and display a square-root-like temperature dependence below $T_C$. Although based on a greatly simplified model, the mean-field simulations give reasonable results for the depolarisation rates: As shown in Fig. 3b, d, f, the magnitude of the slower of the two depolarisation rates obtained by our mean-field description (solid blue line) agrees well with the experimentally determined values of $\lambda_{slow}(T)$ (blue dots) for the smaller nanomagnetic discs (Fig. 3d, f), and the theoretical values only deviate by a factor of two for the largest considered discs with diameter of $d = 70$ nm (Fig. 3b).

The qualitative agreement between the experimentally observed values of $\lambda_{slow}(T)$ and the calculated muon-spin depolarisation indicates that static antiferromagnetic order is the main contribution to $\lambda_{slow}(T)$. The temperature dependence of $\lambda_{fast}(T)$, which peaks at $T_C$, however, is not reproduced by our

simulations since it probably originates from critical correlations emerging at the phase transition, which are not captured by a mean-field description.

Furthermore, it is interesting to note that the measured value of $\lambda_{fast}(T_C)$, seen in Fig. 3, is astonishingly large (up to $15\,\mu s^{-1}$), being more than an order of magnitude larger than $\lambda_{slow}(T)$ (maximum values around $0.7\,\mu s^{-1}$). This is unusual for a $\mu$SR signal, as dynamic depolarisation rates due to a critical slowing down at a phase transition are usually smaller than the depolarisation caused by quasi-static fields[36,38]. Also, the rapid depolarisation described by $\lambda_{fast}(T)$ contributes to the net muon-spin depolarisation with a small fraction $g_{fast}$ only (Table 1), indicating the emergence of a new type of magnetic environment at $T_C$ in a small fraction of the sample. Here, we suggest that the rapid muon-spin depolarisation originates from the critical slowing down in the vicinity of the phase transition, where long-lived correlations can originate from magnetic clusters forming around $T_C$. The relaxation time scales of these correlated regions diverge close to $T_C$, and thus can feature broad, random magnetic field distributions that are static in the $\mu$SR time window, and thus lead to a rapid depolarisation of the muon spin. With the increase of static magnetic order upon further cooling, $\lambda_{slow}(T)$ increases (Fig. 3b, d, f), and the effect of thermal disorder on the muon-spin relaxation, and thus $\lambda_{fast}(T)$ (Fig. 3a, c, e), becomes less relevant.

**Low-temperature blocked regime.** Finally, at low temperatures $T < T_B$ the system is dominated by single-particle blocking, where fluctuations of the individual nanomagnets become much slower than the timescale of the measurement. This behaviour is corroborated by the vanishing fluctuation rate $f_{fluct}$ in dynamic $\mu$SR measurements (Fig. 4e), the saturation of the static magnetic fraction of the muon signal $g_{static}$ (Fig. 4d), and the constant low-temperature value of the slow depolarisation rates $\lambda_{slow}(T)$ and $\lambda_{slow}^{n.i.}(T)$ for strongly and non-interacting samples, respectively (Fig. 3).

As discussed in the introduction, single-particle blocking in artificial XY spin moments only occurs because the magnetic nanodiscs are not perfect, and the value of $T_B$ gives an estimate of the anisotropies present in the system. We find that, across the set of samples considered in this work, the blocking temperature $T_B$ increases as both the disc diameter $d$ as well as the lattice periodicity $a$, decrease (Fig. 3g–i). This can be explained by the fact that the nanofabrication of denser arrays and smaller discs leads to a larger variation in particle size and shape.

For the non-interacting samples, the arrangement of blocked moments is expected to be completely random, governed by the directions of the local anisotropy of the nanoparticles, and the static random local fields lead to a more effective depolarisation of the muon spin than the fluctuating fields at high-temperatures. As moment fluctuations freeze out, the relaxation rate $\lambda_{slow}^{n.i.}(T)$ gradually increases upon cooling, and saturates for $T < T_B$.

To understand the effect of single-particle blocking on the muon-spin relaxation in the strongly interacting samples, we have to consider that magnetic correlations are already present in the vicinity of $T_B < T < T_C$. Here, due to the anisotropy generated from the coupled nanomagnets on the square lattice, and as the moment fluctuations slow down considerably below $T_C$, one can expect that the local moments freeze below $T_B$. However, small anomalies in the relaxation rate $\lambda_{slow}(T)$ (Fig. 3d, f) indicate that the magnetic order becomes slightly less correlated for $T < T_B$. This partial loss of order can be explained by heterogeneous freezing, i.e. a reorientation of the XY moments from the $x$ and $y$ lattice direction favoured in the long-range ordered phase to locally defined random anisotropy axes.

**Emergence of long-range order.** Since low-energy muon-spin relaxation measurements can only establish the presence of a transition to a phase characterized by a local static magnetic field, but not its corresponding long-range order, we have performed auxiliary soft X-ray resonant magnetic scattering measurements at low and high temperatures, with the X-ray energy at the Fe $L_3$ absorption edge. Here, square-lattice artificial dXY spin systems with a lattice periodicity of $a = 120$ nm and disc diameter and disc height of $d = 90$ nm and $h = 5$ nm, respectively, were investigated. The magnetic scattering patterns obtained with linearly polarised X-rays (see Methods for experimental details) display a clear temperature dependence: At high temperature feature-less weak diffuse magnetic scattering intensity is observed (Fig. 5b), whereas at low temperatures streak-like features emerge (Fig. 5a). These features are indicative of ordering, and their streak-like appearance is due to the reflection scattering geometry. The observation of magnetic scattering intensity centred at half-integer positions in reciprocal space at low temperatures corresponds to emergent long-range order that results in a doubling of the structural lattice periodicity, which is compatible with the expected stripe-like ground state of the square-lattice dXY spin system.

## Discussion

Here, we extend the work on artificial spin systems from the study of the thermodynamics of Ising degrees of freedom[18–22] to thermally active magnetic metamaterials with in-plane XY anisotropy. Our results provide robust evidence of a phase transition

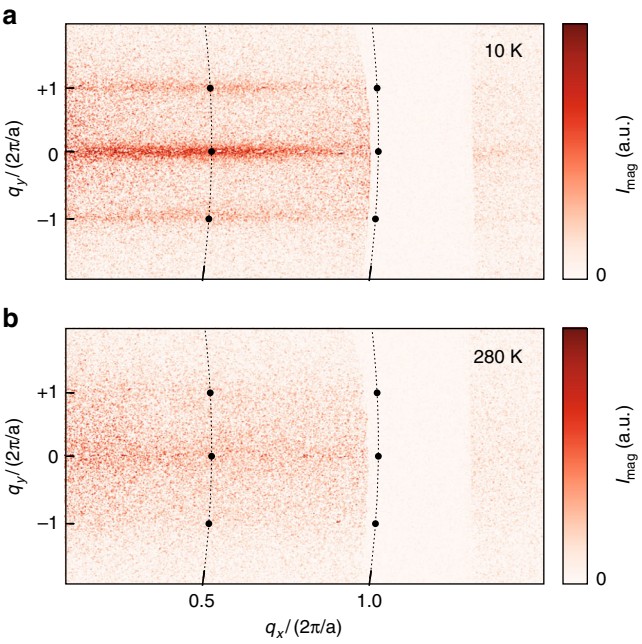

**Fig. 5** Emergent magnetic order in artificial dXY spin system. Using soft X-ray resonant magnetic scattering, we observed emergent correlations in a strongly coupled thermally active square lattice of XY moments. The high-intensity structural Bragg peaks at integer positions are blocked by a mask in order to detect the weaker magnetic signal. **a** At 10 K, the scattered magnetic intensity $I_{mag}$ at half-integer positions in reciprocal space (i.e. corresponding to twice the periodicity of the structural unit cell) originates from magnetic long-range order, and is indicative of the emerging order of the dXY spins. From the fitted peak width, a correlation length of $\lambda_{corr} = 9.4$ unit cells is obtained. **b** At 280 K, no distinct features related to magnetic long-range correlations are discernible in the scattering pattern, as they are destroyed by the rapid superparamagnetic moment fluctuations in the nanoscale discs at high temperatures

as well as long-range order compatible with the theoretically predicted behaviour of a dXY model on a square lattice.

To achieve the continuous in-plane magnetic degrees of freedom, we choose nanoscale permalloy discs that, due to their shape and size, exhibit a macrospin moment confined to the sample plane. The blocking temperature $T_B$ in all investigated samples is at least a factor of two smaller than the transition temperature $T_C$. Therefore, the effect of any imperfections of the discs are small enough, so that at $T_C$ the magnetic moments can still explore the full in-plane angular range and therefore the ensemble is a close approximation of a lattice of XY moments.

Although the magnetostatic interactions between the discs are unlikely to be of purely dipolar character, higher-order multipole contributions are not expected to alter the qualitative zero-field equilibrium behaviour of the square-lattice dXY spin system[11,27]. Our experimental $\mu$SR results, in combination with the soft X-ray scattering measurements, clearly demonstrate that, upon cooling, our strongly coupled artificial XY spin systems develop magnetic correlations and exhibit a phase transition to a long-range ordered state in agreement with predictions for the 2D dXY model.

Finally, as shown by our comparison of three different sample sets, the relevant temperature scales, i.e. the critical temperature $T_C$ and the single-particle blocking temperature $T_B$, can be influenced by careful sample design. Therefore, due to their tunability, where the shape, size, and interdot distances can be precisely tailored, these artificial spin systems enable investigations of ordering phenomena and critical behaviour in the thermodynamic limit of magnetostatically coupled XY systems with controlled amounts of disorder, such as random displacements[39,40] or the presence of vacancies[5,41,42], for which more complex phase diagrams are predicted[43].

## Methods

**Sample fabrication**. The samples were manufactured with electron-beam lithography. Here a polymethyl methacrylate polymer layer was spin-coated on a silicon substrate. Then the desired pattern was exposed with a Vistec EBPG 500Plus electron-beam writer over areas up to 3 cm$^2$. On the developed resist, a thin permalloy (Ni$_{80}$Fe$_{20}$) film was deposited using electron-beam evaporation at base pressure of $3 \times 10^{-7}$ mbar, and capped with a 4 nm layer of gold to prevent oxidation. Following ultrasound-assisted lift-off, the patterned arrays were coated with a continuous 80 nm-thick gold layer, acting as a stopping layer for the muons. SQUID magnetometry confirmed that the patterned nanomagnets support thermally activated moment fluctuations above 70 K. The samples for soft X-ray resonant magnetic scattering experiments were fabricated in a similar manner, and capped with 3 nm Al to avoid oxidation.

**Low-energy muon-spin relaxation**. Experiments were performed at the low-energy muon LEM beamline of the Swiss Muon Source[44,45]. The low-energy (15 keV) spin-polarised muons ($\mu^+$) are implanted into the gold layer, stopping above the arrays of dXY nanomagnets[18]. Here, the muons act as local probes, randomly sampling the local stray fields emanating from the nanomagnets. Stray fields that are transverse to the initial muon spin cause them to precess, and magnetic fluctuations and field distributions lead to a loss of the spin polarisation of the ensemble. The asymmetry of the muon decay leads to an asymmetry between forward- and backward-detected decay positrons $A(t)$, which is related to the (normalised) muon polarisation function via $P(t) = A(t)/A_0$, with the initial asymmetry $A_0$. The relaxation of the depolarisation function $P(t)$ is determined by the variance and fluctuations of the randomly sampled field distribution of the artificial spin system[36].

To characterise the zero-field thermodynamic behaviour of the artificial dXY system, $\mu$SR measurements with the initial muon spin parallel to one of the in-plane lattice directions, i.e. $x$, were performed upon cooling. To decouple the static from the dynamic magnetic response, a longitudinal geometry was used, with the initial spin polarisation parallel to the out-of-plane direction $z$ and a field of 0 mT or 6 mT applied along $z$. At different temperatures between 300 and 5 K, $\mu$SR spectra with $5-30 \times 10^6$ events per time spectrum were measured.

**$\mu$SR fitting procedure**. The model for the zero-field muon-spin polarisation in Eq. (2) considers three contributions to $P(t) = A(t)/A_0$. For the fitting routine, the parameters $A_0$, $g_0$, $g_{slow}$ and $g_{fast}$ are held constant for each sample, and only $\lambda_{slow}$ and $\lambda_{fast}$ are fitted to each measured $P(t, T)$. Fitting was performed with `musrfit`[46] and Python `lmfit`[47]. The value of the initial asymmetry $A_0$ is

determined at high temperatures $T > T_C$. The fraction $g_0$ combines the constant signal due to muons that are not stopped in the sample with the non-relaxing longitudinal contribution (as only transverse field components cause the muon spin to precess) to $P(t)$. The value of $g_0$ is determined in the static magnetic state for $T \ll T_B$. The slow relaxation fraction $g_{slow}$ is the dominant contribution to the muon-spin depolarisation. For non-interacting samples, the fast fraction is not considered, i.e. $g_{fast} \equiv 0$. In contrast, for strongly interacting samples, the value of $g_{fast}$ is determined for temperatures where the rapid depolarisation is most pronounced, i.e. close to $T_C$.

**Mean-field calculation of the muon-spin depolarisation**. We derived the time-dependent $\mu$SR depolarisation function $P(t, \mathbf{x}, T)$ for a muon implanted at position $\mathbf{x}$ by calculating its precession in the static magnetic field generated by the mean-field order parameter $\phi(T)$ of an antiferromagnetic stripe-ordered phase[36]:

$$P(t, \mathbf{x}, T) = \cos^2(\theta(\mathbf{x})) + \sin^2\left(\theta(\mathbf{x})\cos\left(\gamma_\mu |\mathbf{B}(\mathbf{x}, T)| t\right)\right). \quad (3)$$

Here the static field $|\mathbf{B}(\mathbf{x}, T)| \propto \phi(T) \cdot M_{dot}/|\mathbf{x}|^3$ determines the precession dynamics with antiferromagnetic order parameter $\phi(T)$, disc moment $M_{dot}$, and distance $\mathbf{x}$ between muon and XY moment. $\theta = \angle(\mathbf{S}_\mu(0), \mathbf{B})$ denotes the angle between field direction $\mathbf{B}$ and initial muon-spin direction $\mathbf{S}_\mu(0)$, and $\gamma_\mu/(2\pi) = 135.54$ MHz T$^{-1}$ is the muon gyromagnetic ratio. In the experiment, the muon is implanted at random positions. Therefore $P(t, T)$ was averaged over the lateral spread of the muon beam and the implantation depth of the muons in the stopping layer. The latter is weighted by depth probabilities obtained from TRIM.SP depth profiles of muons implanted in an 80 nm gold layer on a silicon substrate[48]. $P(t, T)$ can be fitted by a two-exponential decay, yielding two relaxation rates $\lambda_1(T)$ and $\lambda_2(T)$. Both $\lambda_1(T)$ and $\lambda_2(T)$ follow the square-root temperature dependence of the mean-field order parameter, and the magnitude of the slower relaxation rate compares well with the measured values of $\lambda_{slow}(T)$ (see solid line Fig. 3b, d, f). Further details on the calculations are documented in the Supplementary Note 1.

**Scattering experiments**. Soft X-ray resonant magnetic scattering measurements were performed using the endstation RESOXS[49] at the SIM beamline[50] of the Swiss Light Source, Paul Scherrer Institute. The setup with the small-angle reflection geometry is described elsewhere[19]. In order to be sensitive to the weak magnetic scattering signal, a mask was used to block out the high-intensity structural scattering peaks. To enhance the magnetic signal, the X-ray energy was tuned to the iron $L_3$ absorption edge (708 eV), and the scattering signals collected in the low- and high-temperature phase were measured using a 2D X-ray area detector (Princeton Instruments PME charge-coupled device (CCD) camera with pixel size of 20 μm; sample-detector distance was 175 mm). Scattering patterns obtained with the X-ray energy tuned to 690 eV (a few electron volts below the Fe $L_3$ edge) were used to verify the magnetic origin of the scattered intensity and for normalisation purposes. The magnetic signal shown in Fig. 5 was obtained by subtracting the scattering signals taken with two perpendicular X-ray polarisations, i.e. horizontal and vertical linear polarisations.

**Data availability**. The datasets analysed during the current study are available in the Zenodo repository https://doi.org/10.5281/zenodo.1252365, that also contains a list of the LEM run numbers for which the raw data can be obtained from the SμS data repository (http://musruser.psi.ch).

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

## Acknowledgements

μSR experiments were performed at the LEM (μE4) beamline, Swiss Muon Source SmuS, and soft X-ray scattering experiments were performed at the SIM beamline, Swiss Light Source, both at the Paul Scherrer Institute, Switzerland. This work was funded by the Swiss National Science Foundation (SNSF project grants 200021-155917, 200021-159736, and 200021-172774). D.L. acknowledges funding from the European Union's Horizon 2020 research and innovation programme under the Marie Skłodowska-Curie Grant Agreement No 701647. J.R.L.M. is grateful to the Swiss National Center of Competence in Research, Molecular Ultrafast Science and Technology (NCCR MUST). We thank Vitaly Guzenko and Dario Marty for help with the sample fabrication, Urs Staub for providing support at the SIM beamline, and Peter Holdsworth and Luca Anghinolfi for helpful discussions.

## Author contributions

The experiment was conceived by S.L.L., L.J.H., O.S. and H.L. N.L. fabricated the samples; O.S. assisted with the sample fabrication; N.L., H.L., S.L.L., S.H. and O.S. performed the μSR experiments; A.S., T.P. and Z.S. provided the μSR beamline support; N.L., S.H. and H.L. performed the analysis; D.S., P.M.D. and N.L. performed the simulations; V.S., D.L. and J.L. assisted with the X-ray scattering experiments; N.L., D.S., S.H., P.M.D., H.L., S.L.L. and L.J.H. were involved in the preparation of the manuscript. All authors contributed to the manuscript.
