## [Peer Review File · Nature Communications]

Reviewers' comments:

Reviewer #1 (Remarks to the Author):

The paper reports the results of low-energy muon-spin relaxation measurements on square arrays of magnetic nanodiscs intended to realize a dipolar-coupled 2D XY model. Using a number of samples with different disc diameters and lattice constants the behaviour of the samples is linked to the degree of interaction of the discs. It is found that all samples exhibit a blocking at low temperature but, most significantly, collective magnetic behaviour is identified in an intermediate temperature regime.

This is a very well-written paper reporting interesting results. The main subject of the work - the muon-spin relaxation measurements - are very well described and the data fitting is very professionally handled by well-known experts in this field. The paper should certainly be published in some form. At this point I am less sure about publication in Nature Communications as the case made for the significance of the work is not quite clear in the text. Is this claim that the magnetic ordering that is promoted in the interacting samples is the key result? This presumably should be expected. If, instead, it is the manipulation of the materials, then I would have expected more emphasis on how the growth and structural parameters lead to specific magnetic behaviour (in particular, a detailed discussion of the extent to which that eqn (1) is borne out). Finally, if the claim is that 2D dXY behaviour is engineered, then a far closer link to the specifics of this model would be required. In its current form, although the work is unquestionably of a very high standard, I don't think a good claim is made for its importance in the field. I am therefore not able to recommend publication of the manuscript in its current form. If the authors are able to make a clear case, however, then I would recommend that the paper is given further consideration.

I have a number of other comments on the paper, I make below.

* On page 2 the manuscript the text reads that the "magnetic far field is approximately that of a dipole moment". A more precise statement would be welcome here. Can a length scale be identified for example that relates to the materials in question?

* How can the behavior observed in the LEM experiment be linked more closely to specifics of the 2D dXY model? I accept that a transition to a stripe-ordered state would be expected in such a model, but how can the authors make the link to *this* model in particular. In a mesoscopic system like this, one could imagine that a transition to a quasi-static state might follow for a number of reasons.

* The muSR cannot, in this case, unambiguously identify a transition to a state of LRO. Rather the case is made for a sharp transition of some sort. Can the assignment that the slow relaxation rate is due to static order be justified independently of the simulation?

Why static order, rather than freezing into a disordered state for example? Could the state still host a range of dynamics? Why should an exponential relaxation be attributed to static correlations? Are we in a fast or slow fluctuation limit? More discussion in the main text would be justified here.

* The paper argues on the basis of a mean-field simulations that a transition to LRO is expected and is consistent with the data. However, although the mean-field model seems a perfectly reasonable starting point, arguably the model doesn't provide the conclusive evidence that seems to be argued for here. How robust is the modeling to changes in the underlying spin model, or changes in the assumptions of the mechanism for muon depolarization? There is a lot going on in the description in the supplemental information, but little of this is found in the main text, making it rather difficult to assess this rather important part of the argument.

* The identification of two muon stopping states is certainly justified given the coexistence of signals from two distinct timescales. This is a mesoscopic system, so these states could be quite different. The authors identify clusters or domain walls as two likely culprits. In a microscopic system these would probably exhaust the possibilities, but I wonder if in these materials the large scale nature of the materials allows others (separate freezing behaviour in the discs and in the surrounding areas, for example). I would welcome a comment.

* In terms of basic physics, other stripe-based magnetic structures have been claimed to be rather susceptible to large disorder effects and complicated freezing. Does this result shed any light on those?

In conclusion, this is a very nice paper, full of intriguing physics results which should certainly be published in some form. In its current form it is not entirely clear (at least to me) what the significance of the work is argued to be in the paper, and I would like the authors to have an opportunity to make this case.

Reviewer #2 (Remarks to the Author):

Report on manuscript NCOMMS-17-29314, Collective magnetism in an artificial 2D XY spin system by N. Leo, S. Holenstein, D. Schildknecht et al.

This work presents μ SR measurements of Py nanodisc arrays representing dipolar XY macrospins. Samples with strong and weak dipolar interactions are used to distinguish the signature of the blocking temperature (T_B) of individual nanodiscs from a true collective ordering of the macrospins. The strongly interacting arrays show a temperature range above T_B where the μ SR results clearly reveal a phase transition. Although the characterization of this phase transition is more qualitative than quantitative, I find the results interesting indeed and believe it will inspire further work. The paper is also well written. I can therefore recommend it for publication in Nature Communications.

However, there are a number of issues that need to be considered by the authors before publication.

1) A number of the references in the introduction do not support the statements they are linked to, e.g. [4, 6, 7, 9]. Furthermore, why refer to [13] and [14] when there exist articles clearly stating that the dipolar XY (dXY) system belongs to the 2D Ising class, e.g. PRB 83, 184409 (2011)? Likewise, I cannot see the relevance of [15] and [16], but [17] is sound and [5, 10, 11] also support the view that dXY belongs to the XYh4 class. In addition, it has also been claimed that the dXY transition does not belong to either of the two (Ising, XYh4) [PRB 83, 184409 (2011)].

2) How is T_c determined?

3) page 7, Caption Fig 4 and 3rd paragraph. The authors write $H=0$, 60 G. The cgs unit of magnetic fields is not Gauss, but Oersted. (Although the numbers will be the same.) I also believe that it is better to use SI units, i.e. ($B=$) $\mu_0 H=0$, 6 mT.

4) page 7, Fig 4d). i) A number of data points are missing. Why? ii) I assume that the value of g_{static} is normalized to the maximum value at 5 K. This should be mentioned.

5) page 8, 3rd paragraph, "The inverse relation ... T_c and ... lattice parameter a ..." Not only a , but also the magnetic moment is of importance for the dipolar interaction energy. If one assumes that the magnetization is the same for all samples sets, then Set 2 should have the lowest T_c . An independent measurement (e.g. using SQUID) of the moment or magnetization is therefore needed to link the interaction energy to T_c . The inverse relation between T_c and the lattice parameter might just be coincidence, considering that the magnetization (which can potentially vary with thickness and size) is not taken into account.

6) The discussion about λ_{fast} , page 9-10. One difference between mean field (MF) models and more rigorous treatments of phase transitions is that the former fails to describe fluctuations in the order parameter in the vicinity of T_c . At T_c there will be correlated region of all sizes, while the relaxation time of these regions will diverge (critical slowing down). This will give rise to (very) random local fields and might be the origin of the fast relaxation of the muons and the reason why the MF calculation does not capture this behavior. The concept of domains (and their walls) is not really relevant at T_c , but the term correlated regions captures the physics better.

7) page 10, 4th paragraph. The term surface-to-volume ratio (SV) implies that the whole surface of the disc should be taken into account, but I assume that the authors only consider the perimeter area, where most irregularities should be found. This should be clarified. (Also, if the top area is included in the SV, the trend of the SV is opposite of that of T_B .)

8) Supp. Note, paragraph 3, "... with the magnetization $M=VM_s$ ", should read: "... with the magnetic moment $M=VM_s$ "

9) Supp Note, paragraph 4, $\gamma_{\mu}=135.54$ MHz/T, should read $\gamma_{\mu} / (2\pi)=135.54$ MHz/T

10) The stated value of the gyromagnetic ratio of Py is incorrect (and unphysical, since it corresponds to a g -value below 2.0). A better value of $\gamma_{Py}/(2\pi)$ is 29.5 GHz/T [J.M. Shaw, et al., J. App. Phys. 114, 243906 (2013)].

11) page 3, Eq 6. I do not understand the indices in the equation. A better description of their meaning is needed.

12) page 3, 3rd paragraph, A value for the magnetization of one of the disc sizes is given, but no reference to how this value was determined. Later in the paragraph it is stated that the value $\mu_0 M$

was estimated from T_c . Exactly how? By using $T_c = m^2/a^3 = MV/a^3$?

T_c of a real system is not directly linked to the interaction energy (as in MF model), but is only related to the interaction energy. Determination of the magnetization via T_c is thus a completely unreliable method, and this procedure should only be regarded as a scaling of the MF results to the experimental T_c . The MF calculated value of T_c , using independently determined M , will never give the same value as in the experiment.

Reviewer #3 (Remarks to the Author):

In this manuscript, the authors studied artificial two-dimensional XY spin system with strong dipolar couplings. The most of previously studied artificial dipolar coupled spin systems have Ising anisotropy. Here they studied square-lattice spin systems with XY anisotropy, which show an ordered phase due to dipolar interaction. The physics which is inherent in the dipolar coupled XY spin system is theoretically well studied and well understood from the order by disorder mechanism. I agree that reproducing this phenomenon in experiments has a certain significance. Nevertheless, I think, this paper is not of sufficient quality to merit publication in Nature Communications and I cannot recommend publication of this manuscript. I explain the reason in the following.

From the analysis of muon spin polarization, they showed that there are two characteristic temperatures. This is a plausible and convincing result. They assigned intermediate temperature regime to striped ordered phase. The analysis of this assignment is rather weak. I speculate that other spin configurations can also induce similar increase of depolarization rate. Therefore I don't think they successfully reproduced the order by disorder mechanism.

It is also not clear how strong the XY anisotropy is in this system.

Reviewer #1 (Remarks to the Author):

The paper reports the results of low-energy muon-spin relaxation measurements on square arrays of magnetic nanodiscs intended to realize a dipolar-coupled 2D XY model. Using a number of samples with different disc diameters and lattice constants the behaviour of the samples is linked to the degree of interaction of the discs. It is found that all samples exhibit a blocking at low temperature but, most significantly, collective magnetic behaviour is identified in an intermediate temperature regime.

This is a very well-written paper reporting interesting results. The main subject of the work - the muon-spin relaxation measurements – are very well described and the data fitting is very professionally handled by well-known experts in this field. The paper should certainly be published in some form.

We would like to thank the referee for their interest in our work and their extensive comments. We are happy to hear that our results are considered well written, and do hope that our answers below, and the subsequent changes to our manuscript (in particular the inclusion of Fig. 5 on page 12), make our case more clear.

At this point I am less sure about publication in Nature Communications as the case made for the significance of the work is not quite clear in the text. Is this claim that the magnetic ordering that is promoted in the interacting samples is the key result? This presumably should be expected. If, instead, it is the manipulation of the materials, then I would have expected more emphasis on how the growth and structural parameters lead to specific magnetic behaviour (in particular, a detailed discussion of the extent to which that eqn (1) is borne out). Finally, if the claim is that 2D dXY behaviour is engineered, then a far closer link to the specifics of this model would be required. In its current form, although the work is unquestionably of a very high standard, I don't think a good claim is made for its importance in the field. I am therefore not able to recommend publication of the manuscript in its current form. If the authors are able to make a clear case, however, then I would recommend that the paper is given further consideration.

We thank the referee for pointing this issue out, and we have now improved our claim by first including our recent scattering results, and second by adding further emphasis to the manuscript text.

In this manuscript we demonstrate the experimental realisation of a mesoscopic thermally-active magnetostatically-coupled spin system with continuous degrees of freedom. Here, taking dipolar coupling as the leading-order interaction surely is a simplified approximation. However, theoretical descriptions of such systems yield interesting physics on their own. Our muon-spin relaxation results clearly show the existence of a continuous phase transition in strongly-coupled systems, and the emergent long-range ordered state is further validated by x-ray diffraction measurements, which are now included in the revised manuscript (see Fig. 5 on page 12).

We do believe that these results are important to the field of artificial spin systems, as up to this time the thermal behaviour of Ising-like moments have only been considered. Our results clearly show that also the equilibrium states of moments with continuous degrees of freedom can be readily mimicked, allowing the exploration of the role of geometrical frustration in two-dimensional spin systems

not experimentally explored so far. We consider this a major development in the field of artificial spin systems.

I have a number of other comments on the paper, I make below.

* On page 2 the manuscript the text reads that the "magnetic far field is approximately that of a dipole moment". A more precise statement would be welcome here. Can a length scale be identified for example that relates to the materials in question?

There are two points we wish to consider here:

Firstly, higher-order multipoles (quadrupoles or higher) can indeed contribute to the magnetostatic energy of the thin discs (see Ref. 27), and the approximation of the far field of the magnetic nanodiscs by that of a dipole moment is valid only if the discs are not packed too closely. However, up to a certain extent, one can approach these additional interactions as a correction term added to the dipolar coupling (Ref. 26).

Secondly, and more important for the context of our manuscript, the inclusion of higher-order interactions will not change the character of the symmetry breaking at the phase transition nor affect the expected zero-field stripe-like order (see results of numerical simulations in Ref. 11). Therefore, we believe that our (simplified) point-dipole description of the magnetostatically-coupled moments with continuous spin degrees of freedom captures the main ingredients of the collective behaviour of the system. Further assertions about the exact interaction terms in the experimental system go beyond our current approach.

In the revised manuscript we extend the discussion on the effect of higher-order multipolar interactions on the collective behaviour at the end of the first paragraph on page 2, as well as the first and fifth paragraph on page 3.

* How can the behavior observed in the LEM experiment be linked more closely to specifics of the 2D dXY model? I accept that a transition to a stripe-ordered state would be expected in such a model, but how can the authors make the link to *this* model in particular. In a mesoscopic system like this, one could imagine that a transition to a quasi-static state might follow for a number of reasons.

The referee is correct as low-energy muon spin relaxation spectroscopy cannot unambiguously demonstrate the emergence of a long-range ordered state, but rather the existence of a phase transition only. In order to establish the onset of long-range order, we now include recent results from soft resonant elastic x-ray scattering on artificial dXY spin systems (Fig. 5 on page 12), which clearly show the emergence of magnetic super-structure peaks at low temperatures and establish the onset of long-range correlations.

* The muSR cannot, in this case, unambiguously identify a transition to a state of LRO. Rather the case is made for a sharp transition of some sort. Can the assignment that the slow relaxation rate is due to static order be justified independently of the simulation? Why static order, rather than freezing into a disordered state for example? Could the state still host a range of dynamics? Why should an exponential relaxation be attributed to static correlations? Are we in a fast or slow fluctuation limit? More discussion in the main text would be justified here.

We thank the referee for pointing out these questions. We comment on the different aspects below, and have also added clarifications to our manuscript:

Slow relaxation due to static order: Regarding the association of the slow relaxation to a static moment arrangement, we can confidently point to measurements performed in longitudinal decoupling geometry (Fig. 4), where we find that the magnetic fluctuations considerably slow down below the critical temperature and cease below the single-particle blocking temperature (Fig. 4e). This evolution is in parallel with the growth and saturation of the slow relaxation contribution $\lambda_{\text{slow}}(T)$. We see our mean-field calculations as an additional indication of this assignment.

Exponential muon-spin relaxation due to static magnetic correlations: As the referee probably wishes to point out, in bulk crystals an exponential contribution to the muon-spin relaxation is often associated to spin dynamics. However, even in diluted static spin systems, such as classical spin glasses, exponential muon-spin relaxation can occur (see Ref. 38), and we believe that this scenario is also applicable to our magnetic metamaterial.

Due to the critical slowing down at the transition temperature, the dXY spin system will feature correlated magnetic clusters of diverging size and time scales with random, broad field distributions. At the muon stopping sites, which are distributed laterally and in depth, the field experienced can be well described by a diluted quasi-static random field, leading to an exponential decay of the muon-spin asymmetry.

Fast to slow fluctuation limit: As discussed in the text, we distinguish different regimes for the spin dynamics of the artificial dXY spin systems. At high temperatures (i.e. above T_c) we are clearly in a fast-fluctuating superparamagnetic regime, whereas below T_c our longitudinal measurements indicate that magnetic fluctuations slow down, and cease below the single-particle blocking temperature (see Fig. 4e on page 8). To clarify the different experimental regimes, we added respective statements to the last paragraph on page 9 and the first paragraph on page 10.

Transition to a long-range ordered state: We agree with the referee that the emergence of long-range order cannot be unambiguously proven with a low-energy muon-spin relaxation experiment, as the technique allows the determination of phase transitions to a static magnetic state only. Furthermore, and unlike in bulk crystals, our mesoscopic spin systems do not feature distinct oscillations of the muon-spin asymmetry indicative of specific magnetic environments. This is due to the fact that we average over a wide range of stopping sites with respect to the lattice of the dXY moments distributed in lateral and depth dimensions.

In the revised manuscript, we support our claim of long-range stripe order by complimentary soft x-ray resonant scattering experiments performed on square-lattice artificial dXY spin systems (Fig. 5 on page 12), which clearly show emergent magnetic intensity at half-integer values in reciprocal space. These signatures provide further validation of the assignment of the fast and slow muon-spin relaxation rates to the onset of long-range order in artificial spin systems with continuous degrees of freedom.

* The paper argues on the basis of a mean-field simulations that a transition to LRO is expected and is consistent with the data. However, although the mean-field model seems a

perfectly reasonable starting point, arguably the model doesn't provide the conclusive evidence that seems to be argued for here. How robust is the modeling to changes in the underlying spin model, or changes in the assumptions of the mechanism for muon depolarization? There is a lot going on in the description in the supplemental information, but little of this is found in the main text, making it rather difficult to assess this rather important part of the argument.

We agree with the referee that our mean-field model is based on greatly simplified assumptions and captures the experimentally observed results only partially. Specifically, it cannot capture the dynamic muon-spin relaxation near the transition. While the mean-field model captures the main features of the data, the values of the slow relaxation rates for the largest-diameter dots (Set 1, $d=70$ nm) differ by a factor of two. We therefore decided not to put too much emphasis on the mean-field simulations. We did, however, make small changes in the third paragraph on page 10, and added a reference to the Supplementary Material in the caption to Fig. 3 on page 7.

* The identification of two muon stopping states is certainly justified given the coexistence of signals from two distinct timescales. This is a mesoscopic system, so these states could be quite different. The authors identify clusters or domain walls as two likely culprits. In a microscopic system these would probably exhaust the possibilities, but I wonder if in these materials the large scale nature of the materials allows others (separate freezing behaviour in the discs and in the surrounding areas, for example). I would welcome a comment.

We thank the referee for this comment, but are not quite sure if the question aims at the potential of the discs to exhibit a core-shell structure, or the effect of magnetic moments in the gold stopping layer.

From our recent x-ray scattering results (see Fig. 5 on page 12 added to the revised manuscript) we can say, however, that the length scales of the emergent ordering are associated with the lattice parameter of the dXY spin systems. Therefore, we do believe that our claim relating the slow and fast muon-spin relaxation to specific magnetic environments in our extended artificial metamaterial is valid. As indicated by the mean-field calculations, we can tentatively associate the slow muon-spin relaxation contribution λ_{slow} to the antiferromagnetic order parameter, whereas the fast relaxation contribution most likely originates in the critical slowing down, which leads to highly-correlated clusters in the vicinity of T_C (see modified discussion of the contribution of λ_{fast} at the end of page 10 and the beginning of page 11).

* In terms of basic physics, other stripe-based magnetic structures have been claimed to be rather susceptible to large disorder effects and complicated freezing. Does this result shed any light on those?

The referee is right to point out the disorder, such as vacancies or displacements, can introduce intriguing physical properties and competing phases, such as discussed in the work by Prakash and Henley (Ref. 5), or by Pastor and Jensen (Refs. 40 and 41). Furthermore, a recent theoretical assessment of the phase diagram of dXY spin systems shows that vortex-like spin configurations are even more easily stabilised by small amounts of lattice disorder than previously thought (D. Schildknecht *et al.*, in preparation).

Our work on thermally-fluctuating strongly-interacting dXY spin systems opens indeed a door for future investigations of such disordered systems.

In conclusion, this is a very nice paper, full of intriguing physics results which should certainly be published in some form. In its current form it is not entirely clear (at least to me) what the significance of the work is argued to be in the paper, and I would like the authors to have an opportunity to make this case.

We again thank the referee for the insightful comments and hope that our modifications to the manuscript, as well as our answers to the referee's questions, make our case more clear.

Reviewer #2 (Remarks to the Author):

Report on manuscript NCOMMS-17-29314, Collective magnetism in an artificial 2D XY spin system by N. Leo, S. Hohenstein, D. Schildknecht et al.

This work presents μ SR measurements of Py nanodisc arrays representing dipolar XY macrospins. Samples with strong and weak dipolar interactions are used to distinguish the signature of the blocking temperature (T_B) of individual nanodiscs from a true collective ordering of the macrospins. The strongly interacting arrays show a temperature range above T_B where the μ SR results clearly reveal a phase transition. Although the characterization of this phase transition is more qualitative than quantitative, I find the results interesting indeed and believe it will inspire further work. The paper is also well written. I can therefore recommend it for publication in Nature Communications.

However, there are a number of issues that need to be considered by the authors before publication.

We are delighted that the referee finds our results interesting and well-presented, and suitable for publication in Nature Communications. We thank them for the extensive questions and comments and hope that our answers, and the changes to the manuscript, make our case more convincing.

1) A number of the references in the introduction do not support the statements they are linked to, e.g. [4, 6, 7, 9]. Furthermore, why refer to [13] and [14] when there exist articles clearly stating that the dipolar XY (dXY) system belongs to the 2D Ising class, e.g. PRB 83, 184409 (2011)? Likewise, I cannot see the relevance of [15] and [16], but [17] is sound and [5, 10, 11] also support the view that dXY belongs to the XYh4 class. In addition, it has also been claimed that the dXY transition does not belong to either of the two (Ising, XYh4) [PRB 83, 184409 (2011)].

We thank the referee for pointing out potential mistakes in the reference assignment. After checking the literature again we decided to remove Ref. 4 (Cardy & Ostlund, PRB 25 6899, 1982), Ref. 13 (Lee & Teitel, PRB 46 3247, 1992), Ref. 14 (Lapilli *et al.*, PRL 96 140603, 2006), and Ref. 16 (Berthier *et al.*, J. Phys. A 34 1805, 2001) in our revised manuscript.

We also wish to thank the referee for pointing out the paper by Beak *et al.* (PRB 83 184409, 2011, now Ref. 12), which we mistakenly forgot to cite in our previous

version. The changes were all applied to the first paragraph on page 2, where we shifted some references to better indicate the respective statements they are supporting (all changes are marked in red).

As the universality class of the phase transition in the square-lattice dXY system has not been theoretically or experimentally resolved yet, we do believe that the works by De'Bell *et al.* (PRB **55** 15108, 1997) and Carbognani *et al.* (PRB **62** 1015, 2000) contribute to the discussion, and have decided to leave them in the manuscript (Refs. 6 and 7 in the revised manuscript).

2) How is T_c determined?

We determined the value of T_c by a piecewise linear fit $1/\lambda_{\text{fast}} \sim (T - T_c)$ to the apparent divergence of a λ_{fast} , as well as a piecewise linear fit to the values of λ_{slow} . Here, the first option yielded results with a better confidence interval, which is the value reported in Tab 1.

We added this detail to the second paragraph on page 6.

3) page 7, Caption Fig 4 and 3rd paragraph. The authors write $H=0$, 60 G. The cgs unit of magnetic fields is not Gauss, but Oersted. (Although the numbers will be the same.) I also believe that it is better to use SI units, i.e. ($B=$) $\mu_0 H=0$, 6 mT.

We thank the referee for this comment, and adapted the magnetic field units in Fig. 4 and its the caption, as well as the first paragraph on page 8.

4) page 7, Fig 4d). i) A number of data points are missing. Why? ii) I assume that the value of g_{static} is normalized to the maximum value at 5 K. This should be mentioned.

i) We thank the referee for pointing out this apparent missing data. The data points in Fig. 4d are obtained by integrating the area between the muon-spin relaxation curves at zero field and 6 mT. In contrast, in Figs. 4e,f additional data points taken at 6 mT only are shown as well. Without comparative zero-field measurement the fits are less constrained and therefore the fit parameters are less reliably determined.

We added a sentence (marked in red) to the end of the first paragraph on page 8 to clarify the discrepancy between the number of data points in Figs. 4d-f.

ii) Yes, we did normalise the area between the longitudinal muSR measurements, which we denote as static fraction g_{static} , to the low-temperature value at 10 K.

We added a respective note (red) in the caption of Fig. 4 (page 8) to clarify this.

5) page 8, 3rd paragraph, "The inverse relation ... T_c and ... lattice parameter a ..." Not only a , but also the magnetic moment is of importance for the dipolar interaction energy. If one assumes that the magnetization is the same for all samples sets, then Set 2 should have the lowest T_c . An independent measurement (e.g. using SQUID) of the moment or magnetization is therefore needed to link the interaction energy to T_c . The inverse relation between T_c and the lattice parameter might just be coincidence, considering that the magnetization (which can potentially vary with thickness and size) is not taken into account.

We agree that here the term "inverse relationship" might be too strong here, as the dXY moment magnitude, determined by disc volume and saturation magnetisation, is not separately accounted for. Nevertheless, a reduction of lattice parameter is

expected to have a strong effect on the transition temperature, since the lattice periodicity a appears in the third power in Eq. (1).

We accordingly shortened the text in the second paragraph on page 9.

We also found that the saturation magnetisation is *not* the same for all the sample sets, either due to a reduction of moment in the thinner films, or oxidation of discs with a larger surface area.

6) The discussion about λ_{fast} , page 9-10. One difference between mean field (MF) models and more rigorous treatments of phase transitions is that the former fails to describe fluctuations in the order parameter in the vicinity of T_c . At T_c there will be correlated region of all sizes, while the relaxation time of these regions will diverge (critical slowing down). This will give rise to (very) random local fields and might be the origin of the fast relaxation of the muons and the reason why the MF calculation does not capture this behavior. The concept of domains (and their walls) is not really relevant at T_c , but the term correlated regions captures the physics better.

As the referee correctly points out the (slow) fluctuations at the critical temperature are the most likely culprit for the random fields that lead to the observed rapid muon-spin depolarisation, and that such correlated fluctuations are not captured within a mean-field description. The concept of domain walls appeared from our discussion as an alternative scenario since domain walls are (mobile) disordered regions with highly anisotropic local fields, but without extended simulations we cannot test this (rather unfavourable) scenario.

We realised that our original discussion was somewhat unclear, and have therefore decided to reduce the discussion on the origin of the fast muon-spin relaxation contribution to the effect of correlated fluctuations in the vicinity of the critical phase transition temperature – see end of first paragraph on page 11.

7) page 10, 4th paragraph. The term surface-to-volume ratio (SV) implies that the whole surface of the disc should be taken into account, but I assume that the authors only consider the perimeter area, where most irregularities should be found. This should be clarified. (Also, if the top area is included in the SV, the trend of the SV is opposite of that of T_B .)

We thank the referee for pointing out the inaccurate language. Here, we do consider the full surface area of the disc, i.e. circular top and bottom areas plus the disc perimeter times the disc height. Using this definition, we do find an apparent linear relationship between the blocking temperature and the surface-to-volume ratio. As the referee correctly points out, this relationship is inverse to what we wrote in our text, as discs with a smaller surface-to-volume ratio exhibit higher blocking temperatures.

Although this is not a very strong claim, we wish to point out that the superparamagnetic blocking of circular discs might follow a different dependency on structural parameters (e.g. the surface-to-volume ratio as suggested in our experiments) compared to Ising moments (where $T_B \sim \text{volume}$).

We do not know the reason for this behaviour, and the small number of three data points does not allow the determination of a systematic relationship. After some

consideration, we therefore decided to remove the respective passage from the text (see shorted third paragraph on page 11).

8) Supp. Note, paragraph 3, "... with the magnetization $M=VMs$ ", should read: "... with the magnetic moment $M=VMs$ "

We thank the referee for noting this inaccurate choice of words and changed the text accordingly (marked red on page 1 of the Supplementary Note).

9) Supp Note, paragraph 4, $\gamma_{\mu}=135.54$ MHz/T, should read $\gamma_{\mu} / (2\pi)=135.54$ MHz/T

We thank the referee for pointing out the correct prefactor, which we use in the current Supplementary Note (marked red in last paragraph of page 1). The validity values for our approximation were re-calculated with this value (see first paragraph on page 2 for updated percentages).

10) The stated value of the gyromagnetic ratio of Py is incorrect (and unphysical, since it corresponds to a g-value below 2.0). A better value of $\gamma_{Py}/(2\pi)$ is 29.5 GHz/T [J.M. Shaw, et al., J. App. Phys. 114, 243906 (2013)].

We thank the referee for pointing out the gyromagnetic ratio of thin-film permalloy, and used the value of the given reference to re-calculate the validity criterion of our mean-field approach, as indicated in the first paragraph of page 2 of the Supplementary Note.

The previous percentage for the validity of our mean-field assumption for the dXY lattices with 55 nm, 70 nm, and 100 nm lattice periodicity were 87%, 81%, and 70%, respectively. In contrast, the re-calculated values are lower with 71%, 60%, and 40%. Although these values are somewhat lower, our mean-field approximation is still justified for the smaller-periodicity lattices (55 nm and 70 nm), as the comparison to the experiment yields a good agreement (see comparison in Figs. 3d,f on page 7).

11) page 3, Eq 6. I do not understand the indices in the equation. A better description of their meaning is needed.

We thank the referee for this comment. Indeed, we used different notations for the indices in the Supplementary equations (4) and (6), and rectified this discrepancy by adapting the indices in Eqs. (4) and (5) on page 2 and 3 – changes are marked in red.

12) page 3, 3rd paragraph, A value for the magnetization of one of the disc sizes is given, but no reference to how this value was determined. Later in the paragraph it is stated that the value $\mu_0 M$ was estimated from T_c . Exactly how? By using $T_c=m^2/a^3=MV/a^3$? T_c of a real system is not directly linked to the interaction energy (as in MF model), but is only related to the interaction energy. Determination of the magnetization via T_c is thus a completely unreliable method, and this procedure should only be regarded as a scaling of the MF results to the experimental T_c . The MF calculated value of T_c , using independently determined M , will never give the same value as in the experiment.

In point-dipole approximation the proportionality between T_c and m^2/a^3 , as stated in Eq. (1) on page 3, holds, and we agree that for extended dots or higher-order multipole interactions this might only be a rough approximation of the transition

temperature of a “real” system. However, based on our results we cannot make any conclusive statement on the required correction terms.

As the referee correctly points out, we are using the disc volume and the experimentally-measured T_c to infer the volume magnetisation M of Sample Sets 1 and 3 by comparing it to the result for Sample Set 2. This value of M is then used to re-scale the calculated muon-spin relaxation. Experimentally, we can determine the value of the disc magnetisation via SQUID magnetometry. However, due to their small mass and spurious background contributions, we found it difficult to obtain reliable measures of magnetisation values, and are confident only for the magnetisation values of samples of Sample Set 2. The reduced volume magnetisation of our nanostructures, compared to the bulk magnetisation of permalloy, is in line with published results of other groups. Also, our observations indicate a thickness-dependent change in the volume magnetisation, which always needs to be taken into account for the design of thermally-active artificial spin systems. We clarify these points in the second-to-last paragraph on page 3 of the Supplementary Note.

Reviewer #3 (Remarks to the Author):

In this manuscript, the authors studied artificial two-dimensional XY spin system with strong dipolar couplings. The most of previously studied artificial dipolar coupled spin systems have Ising anisotropy. Here they studied square-lattice spin systems with XY anisotropy, which show an ordered phase due to dipolar interaction. The physics which is inherent in the dipolar coupled XY spin system is theoretically well studied and well understood from the order by disorder mechanism. I agree that reproducing this phenomenon in experiments has a certain significance. Nevertheless, I think, this paper is not of sufficient quality to merit publication in Nature Communications and I cannot recommend publication of this manuscript. I explain the reason in the following.

We thank the referee for their interest in our paper and do hope that our answers below and the modifications made to our manuscript will lead to a more favourable impression of the significance of our results.

From the analysis of muon spin polarization, they showed that there are two characteristic temperatures. This is a plausible and convincing result. They assigned intermediate temperature regime to striped ordered phase. The analysis of this assignment is rather weak. I speculate that other spin configurations can also induce similar increase of depolarization rate. Therefore I don't think they successfully reproduced the order by disorder mechanism.

We do agree that the experimental signatures of the muon-spin relaxation can be linked to the presence of a phase transition only, while the stripe-ordered nature of the phase was inferred on the relatively weak grounds of our mean-field approach.

In our current manuscript version, we now include recently obtained soft resonant elastic x-ray scattering patterns from artificial dXY spin systems (see Fig. 5 and the discussion on page 12 in the revised manuscript), where magnetic scattering peaks at half-integer values unambiguously demonstrate the onset of long-range order at low temperatures. We are therefore confident that our association of the measured

muon-spin relaxation contributions to ordered moment configurations is a valid claim.

It is also not clear how strong the XY anisotropy is in this system.

The referee asks an interesting question, for which we have two interpretations:

On one hand, we can understand the term “XY anisotropy” with respect to the ratio of in-plane vs. out-of-plane moment. For our samples, the choice of material (with very little magneto-crystalline anisotropy) and the shape anisotropy, i.e. diameter-to-thickness ratio, of the magnetic discs dictate a very strong anisotropy for the moment to be planar.

On another hand, the term “XY isotropy” might refer to the isotropy of the in-plane moment, i.e. how well the disc’s magnetisation can be described by a proper rotor-like moment. Here, we would like to point out that the in-plane anisotropy of the individual dots (e.g. caused by irregular disc shapes) can be linked to the experimentally-determined single-particle blocking temperatures T_B . In the strongly-interacting samples, the value of T_B (given by the shape anisotropy) is at least a factor of two lower than the critical temperature T_C caused by the magnetostatic coupling between the nanodiscs on the square lattice (i.e. interaction anisotropy). Therefore, we do believe that the thermally-fluctuating moments at T_C exhibit, to a good approximation, a continuous in-plane anisotropy, and thus allow us to study the equilibrium properties of artificial XY spin systems.

We further elaborate on these points in the manuscript Discussion (page 13f).

REVIEWERS' COMMENTS:

Reviewer #1 (Remarks to the Author):

I have read the authors' response to all of the reviewer questions, along with the revised manuscript.

The authors have done a very thorough job indeed in considering all of the referee queries and criticisms, have answered them fully and revised the manuscript quite extensively. Especially welcome is the inclusion of new x-ray scattering data, which are helpful in clarifying the claims made about long-range correlations, which were previously rather ambiguous. The authors have also updated (and in many cases simplified) the text and provided a clearer motivation and context for the study and its importance.

As the authors have done everything I suggested in my review and, as far as I can tell, have provided satisfactory answers to the other referees, I am happy to recommend the paper for publication in Nature Communications, for all of the reasons given in my original report.

Reviewer #2 (Remarks to the Author):

The axes of Figure 5 need labels. Apart from this remark I now find the manuscript ready for publication in Nature Communications. The authors have provided good replies to the referees' comments and have improved the paper accordingly. In addition, the new x-ray results strengthen the report and clearly reveal a low-temperature ordered state.

Reviewer #3 (Remarks to the Author):

The authors improved the manuscript by appropriate and careful revisions. They added a new nice result (Fig.5) of soft resonant elastic X-ray scattering, which gives a direct evidence of stripe-ordered spin configurations. Now I agree that the authors succeeded to show order-by-disorder mechanism in artificial spin systems. I can recommend publication of this manuscript in Nature Communications.

Reviewer #1 (Remarks to the Author):

I have read the authors' response to all of the reviewer questions, along with the revised manuscript.

The authors have done a very thorough job indeed in considering all of the referee queries and criticisms, have answered them fully and revised the manuscript quite extensively. Especially welcome is the inclusion of new x-ray scattering data, which are helpful in clarifying the claims made about long-range correlations, which were previously rather ambiguous. The authors have also updated (and in many cases simplified) the text and provided a clearer motivation and context for the study and its importance.

As the authors have done everything I suggested in my review and, as far as I can tell, have provided satisfactory answers to the other referees, I am happy to recommend the paper for publication in Nature Communications, for all of the reasons given in my original report.

Reviewer #2 (Remarks to the Author):

The axes of Figure 5 need labels. Apart from this remark I now find the manuscript ready for publication in Nature Communications. The authors have provided good replies to the referees' comments and have improved the paper accordingly. In addition, the new x-ray results strengthen the report and clearly reveal a low-temperature ordered state.

Reviewer #3 (Remarks to the Author):

The authors improved the manuscript by appropriate and careful revisions. They added a new nice result (Fig. 5) of soft resonant elastic X-ray scattering, which gives a direct evidence of stripe-ordered spin configurations. Now I agree that the authors succeeded to show order-by-disorder mechanism in artificial spin systems. I can recommend publication of this manuscript in Nature Communications.

We thank the referees for their positive comments, and are delighted that the changes to our manuscript were well received. As asked by Referee #2, we now added axis labels to the scattering patterns in Fig. 5.